# New Insights into the Crosstalk among the Interferon and Inflammatory Signaling Pathways in Response to Viral Infections: Defense or Homeostasis

**DOI:** 10.3390/v14122798

**Published:** 2022-12-15

**Authors:** Jingwen Dai, Pingping Zhou, Su Li, Hua-Ji Qiu

**Affiliations:** 1State Key Laboratory of Veterinary Biotechnology, Harbin Veterinary Research Institute, Chinese Academy of Agricultural Sciences, Harbin 150069, China; 2Department of Immunology, School of Basic Medicine, Harbin Medical University, Harbin 150081, China

**Keywords:** crosstalk, viral infections, interferon signaling, inflammatory response, homeostasis

## Abstract

Innate immunity plays critical roles in eliminating viral infections, healing an injury, and restoring tissue homeostasis. The signaling pathways of innate immunity, including interferons (IFNs), nuclear factor kappa B (NF-κB), and inflammasome responses, are activated upon viral infections. Crosstalk and interplay among signaling pathways are involved in the complex regulation of antiviral activity and homeostasis. To date, accumulating evidence has demonstrated that NF-κB or inflammasome signaling exhibits regulatory effects on IFN signaling. In addition, several adaptors participate in the crosstalk between IFNs and the inflammatory response. Furthermore, the key adaptors in innate immune signaling pathways or the downstream cytokines can modulate the activation of other signaling pathways, leading to excessive inflammatory responses or insufficient antiviral effects, which further results in tissue injury. This review focuses on the crosstalk between IFN and inflammatory signaling to regulate defense and homeostasis. A deeper understanding of the functional aspects of the crosstalk of innate immunity facilitates the development of targeted treatments for imbalanced homeostasis.

## 1. Introduction

The immune system is the multifunctional defense system of host cells. The host generally maintains homeostasis by modulating immune responses. Innate immunity is the first line of defense against viral infections and includes natural physiological barriers and innate immune responses.

In innate immune responses, pattern recognition receptors (PRRs) recognize the conserved components of the virus [1] and trigger the activation of the interferon (IFN) [2,3], nuclear factor kappa B (NF-κB) [4,5], and inflammasome signaling pathways [6,7]. There is complex crosstalk or interplay among the signaling pathways. The recognition of intracellular pathogen-associated molecular patterns (PAMPs) by PRRs initiates the NF-κB signaling pathway, which ultimately triggers the transcription and expression of proinflammatory cytokines. Moreover, the activation of the NF-κB signaling pathway may facilitate the transcription of IFNs. In addition, the activation of NF-κB promotes the transcription of the pro-interleukin (pro-IL)-1β and pro-IL-18, and the production of mature IL-1β and IL-18 is regulated by the inflammasome signaling. The inflammasome consists of multimeric proteins, including the receptor proteins apoptosis-associated speckle-like protein (ASC) and caspase-1. Once formed, the inflammasome activates caspase-1, which proteolytically activates the proinflammatory cytokines IL-1β and IL-18 [8]. Subsequently, inflammasome activation causes pyroptosis, which is a rapid pro-inflammatory state of cell death [9]. In addition, IFN activates the Janus kinase (JAK)/signal transducer and activator of transcription (STAT) signaling pathway, which enables the host to generate several IFN-stimulated genes (ISGs) to elicit antiviral effects [10,11,12,13].

Many innate immune molecules are involved in the signal transduction of the inflammatory and IFN signaling pathways, some of which participate in the activation of multiple pathways. For example, the inhibitor of NF-κB kinase trimer (IKK) can activate both the IFN and NF-κB signaling pathways [14,15]; protein kinase R(PKR) is expressed downstream in the IFN signaling pathway, leading to a reduction in protein translation and inhibitor of NF-κB (IκB) degradation, which triggers the activation of the NF-κB signaling pathway. The innate immune signaling pathways comprise a vast network of information transmission in the host, and these pathways are not independent but interact with each other in some way (Figure 1). Notably, the imbalance in the crosstalk between the innate immune responses usually causes tissue injury in the host. For example, Zika virus (ZIKV) can leverage the crosstalk between the IFN and inflammasome pathways to cause the excessive activation of inflammasomes, which results in immunologic injury.

The crosstalk between these signaling pathways exhibits advantageous or adverse effects on the host, and new antiviral strategies can be developed by identifying the targets in the crosstalk of the signaling pathways.

## 2. Activation of the IFN and Downstream Signaling Pathways in Response to Viral Infections

The IFN signaling pathway is a major component of innate immunity and plays important roles in host antiviral immunity. Viral infection triggers the production of IFNs and activates downstream pathways to initiate the expression of ISGs.

### 2.1. IFNs and IFN-Stimulated Genes

IFNs are small proteins that are produced and secreted into cell supernatants in vitro and secreted into the extracellular matrix in vivo. PRRs recognize PAMPs in cells and then trigger the interferon regulatory factor 3 (IRF3) and NF-κB signaling pathways; ultimately, the generated IFN is secreted into cell supernatants. During RNA virus infections, retinoic acid-inducible gene-I (RIG-I) and melanoma differentiation-associated gene 5 (MDA5), the members of the RIG-I-like receptor (RLR) family, recognize viral nucleic acids and activate the caspase recruitment and activation domain (CARD) functional region of mitochondrial antiviral signaling protein (MAVS) on mitochondria, and the activated CARD triggers the NF-κB signaling pathway and IRF3 nuclear translocation to initiate IFN transcription [16,17,18] (Figure 1). IFN exerts its biological effects by binding to target cellular membrane receptors, which stimulates the activation of the JAK family of tyrosine kinases. Subsequently, transcription factors are activated by tyrosine phosphorylation and act as signal transduction factors and transcriptional activators. IFNs are divided into three classes, type I, II, and III IFNs [19,20], depending on the membrane receptors to which they bind. IFNs exert antiviral activities via the JAK/STAT signaling pathway.

Type I IFNs are important components of innate immunity and exert antiviral effects. Type I IFNs include IFN-α, IFN-β, IFN-ε, IFN-κ, and IFN-ω [21]. IFN binds to relevant receptors to activate the kinases that phosphorylate tyrosine residues of the STAT family members. The recognition receptor for type I IFN is interferon-α/β receptor (IFNAR), which is a heterodimer consisting of two subunits, IFNAR1 and IFNAR2 [22], and is widely expressed in a variety of cells. After binding to IFNARs, IFNAR1 and IFNAR2 recruit tyrosine kinase 2 (TYK2) and JAK1, respectively, and trigger the immune response signaling process [23]. Type I IFN-activated STAT1 and STAT2 bind to IRF9 to form the nuclear translocation complex called interferon-stimulated gene factor 3 (ISGF3) [24], which is translocated into the nucleus, where it binds to interferon stimulation response element (ISRE) sequences to initiate the transcription of several ISGs [25].

Type II IFN is an important cytokine that is involved in the regulation of the immune response, especially cellular immunity. IFN-γ is the only type II IFN and is mainly expressed by activated natural killer (NK), NKT, and T cells [26,27,28,29,30,31,32,33,34]. Interferon-gamma receptor (IFNGR), the type II IFN receptor, consists of IFNGR1 and IFNGR2 subunits and is widely expressed in a variety of cell types. Upon type II IFN binding to IFNGR, IFNGR1 binds to JAK1, and IFNGR2 binds to JAK2 [35]. Then, the JAKs are activated through phosphorylation, leading to STAT1 phosphorylation, and the phosphorylated STAT1 forms a homodimer that binds the gamma-activating sequence (GAS) in DNA to initiate ISG transcription and functions in the regulation of immune responses [36].

Type III IFNs exert tissue-specific effects and are associated with allergic reactions and autoimmune diseases. The type III IFN family in humans includes four subtypes, IFN-λ1 (IL-29), IFN-λ2 (IL-28A), IFN-λ3 (IL-28B), and IFN-λ4 [37]. However, the type III IFN family in mice consists of IFN-λ2 and IFN-λ3 [38]. The receptor of type III IFNs is a heterodimer consisting of IL-10R2 and IL-28Rα, and it is expressed preferentially on epithelial cells, as well as neutrophils [39]. Type III IFNs also interact with a unique heterodimer consisting of IFN-λR1 and with a unique heterodimer composed of IFN-λR1 and IL-10R2. Despite their different receptors, the downstream signaling pathways and transcriptional responses activated by types I and III IFNs exhibit substantial overlap. Both types I and III IFNs signal through the JAK-STAT pathway to activate the heterotrimeric transcription factor complex ISGF3 [40]. Upon binding to IFN-λ, IFN-λR1 activates JAK1, and IL-10R2 activates TYK2, initiating the JAK/STAT signaling pathway [41]. Usually, mitochondria-localized MAVS induces an antiviral response typified by the expression of type I IFNs and ISGs. In contrast, RLR signaling via MAVS on peroxisomes does not induce the expression of type I IFNs but does induce the expression of ISGs [42]. It has been shown that MAVS on peroxisomes induces secretion of IFN-λ in human cells (especially in epithelial cells) that can cross species to activate a JAK/STAT pathway regulated by JAK2, and MAVS on peroxisomes induces selective expression of IFN-λ, but not of type I IFNs [43], indicating that RLRs induce type III IFN expression in a variety of human cell types (Figure 2).

### 2.2. ISGs Exert Antiviral Effects

IFNs trigger the activation of the JAK-STAT signaling pathway, leading to the expression of hundreds of ISGs that can be divided into antiviral effectors and negative or positive regulators of the IFN signaling pathway [36].

#### 2.2.1. Antiviral Effectors

ISGs targeting DNA viral infection in transcriptional proteins include cyclic GMP-AMP synthase (cGAS) and absent in melanoma 2 (AIM2)-like receptors (ALRs). cGAS is a receptor for DNA viruses, and retroviruses such as human immunodeficiency virus (HIV) are involved in the cGAS/stimulator of interferon genes (STING) pathway to induce type I IFNs and other cytokines [44,45]. ALRs mainly recognize intracellular or nuclear pathogenic DNA or abnormal cellular DNA; on the one hand, they activate the cGAS/STING signaling pathway to initiate type I IFN transcription by activating TANK-binding kinase 1 (TBK1) to activate IRF3/7 and on the other hand, they form inflammasomes to induce the inflammatory response [46].

ISGs targeting RNA virus infections include oligoadenylate synthetases/ribonuclease L (OAS/RNase L), viperin, tetherin, cholesterol 25-hydroxylase (CH25H), interferon-induced transmembrane protein (IFITM), tripartite motif-containing protein (TRIM) and RLRs. OAS/RNase L can recognize foreign RNA and block viral infection by cleaving viral and cellular single-stranded RNA [47]. Viperin targets the nonstructural (NS)3 protein of ZIKV or tick-borne encephalitis virus (TBEV) for proteasomal degradation, inhibiting viral replication, and it binds to STING to enhance the activation of the type I IFN pathway in human embryonic kidney 293T cells (HEK293T) and HeLa cells [48,49]. Tetherin is a type II transmembrane protein with a unique topology that regulates the host response to viral infection by inhibiting the release of progeny virions. Meanwhile, tetherin exerts an inhibitory effect on the replication of enveloped viruses in vitro, such as influenza A virus (IAV), dengue virus (DENV), Ebola virus (EBOV) in HEK293T and HeLa cells, human immunodeficiency virus (HIV) and respiratory syncytial virus (RSV), which are sensitive to the antiviral activity of tetherin [50,51]. The CH25H protein is a hydroxylase capable of inhibiting the replication of severe acute respiratory syndrome coronavirus 2 (SARS-CoV-2) in HEK293T and Vero cells [52] and other coronaviruses by blocking membrane fusion in human lung epithelial cells [53]. IFITM is a transmembrane protein that is involved in the regulation of CD4^+^ T helper cell differentiation in a T-cell-intrinsic manner [54]. In addition, IFITM can inhibit the infectivity of various viruses [55]. For example, IFITM inhibits SARS-CoV-2 infection; While IFITM3 inhibits endocytic entry of the virus, it enhances virus fusion at the plasma membrane in HEK293T cells [56]. TRIM proteins are primarily E3 ubiquitin ligases [57], TRIM25 mediates the activation of the RIG-I signaling pathway in the antiviral response in HEK293T cells [58], whereas TRIM22 can enhance the JAK-STAT1/2 signaling pathway to inhibit RSV replication in human epithelioma-2 (HEP-2) cells [59].

RLRs, as PRRs of RNA viruses, mainly recognize double-stranded RNA of viruses [60,61,62], thereby initiating downstream innate immune signaling pathways and activating IFN transcription through the RIG-Ι/MDA-5 signaling pathway [63,64].

The myxovirus resistance protein (Mx) mainly impairs the infection of RNA viruses; for example, the mouse Mx1 inhibits RNA virus infections in mice, such as influenza virus (IV) [65,66] and Thogoto virus (THOV) [67], while the human MxA can reduce the infectivity of RNA viruses, such as infectious bursal disease virus (IBDV) and mammalian reovirus (MRV) in Vero cells [68]. MxA also can reduce the infection of DNA viruses, such as the African swine fever virus (ASFV) infection in Vero and porcine kidney (PK)-15 cells [69] (Table 1).

#### 2.2.2. Positive Regulators of the IFN Signaling Pathway

The positive regulators include STAT1/2, RLRs, ALRs, cGAS, OAS/RNase L, PKR, IRF1, IRF9 [70], IRF3 [71], and IRF7 [72]. PKR induces a significant degradation of IκBα in host cells, which subsequently triggers the activation of the NF-κB signaling pathway [73]. RLRs and ALRs act as PPRs and initiate the transcription of type I IFN upon recognition of the corresponding ligands. STAT1/2, cGAS, and IRF1, 3, 7, and 9 function as the adaptors or transcription factors of innate immunity, which play critical roles in the activation of the NF-κB signaling pathway and cGAS/STING and RIG-Ι/MDA-5 signaling pathways, respectively, thus playing a positive role in the regulation of the IFN signaling pathway in response to viral infections. A previous study showed that TRIM26 promotes the interaction of TBK1 with NF-κB essential modulator (NEMO), leading to the activation of the IFN signaling in HEK293T and HeLa cells [74]. TRIM26 positively regulates cytoplasmic dsRNA-induced expression of type I IFNs by facilitating TBK1-NEMO interaction (Table 1).

#### 2.2.3. Negative Regulators of the IFN Signaling Pathway

The negative regulators include suppressor of cytokine signaling (SOCS), ubiquitin specific peptidase 18 (USP18), and TRIM26. In human cells, the SOCS protein inhibits activation of the JAK/STAT signaling pathway by binding to the phosphorylated tyrosine residues in the IFN receptor or JAK protein, thereby inhibiting STAT binding and JAK activity and exerting an inhibitory effect on the IFN signaling pathway, which can also be shown in mice [75,76]. USP18, the first identified ISG15-specific protease, can inhibit the expression of ISGs in mammalian cells [77,78,79]. USP18 specifically binds to the IFNAR2 receptor subunit and inhibits the activity of receptor-associated JAK1 by blocking the interaction between JAKs and the IFN receptor, thereby inhibiting the activation of the IFN signaling pathway [80].

IRF3 activation requires phosphorylation, dimerization, and nuclear translocation while the IFN pathway is activated. Interestingly, as a positive regulator of the IFN signaling pathway, TRIM26 exerts an inhibition effect on IFN production. TRIM26 in humans or mice binds to IRF3 and promotes its K48-linked polyubiquitination and degradation in the nucleus, thereby inhibiting the IFN-β signaling [81] (Table 1).

**Table 1 viruses-14-02798-t001:** Functions of ISGs-encoded proteins.

Proteins	Targets	Functions	Modes	References
Mx	DNA or RNA viruses	Mouse Mx1 inhibits RNA virus infections, such as IV and THOV	Antiviral effectors	[65,67]
		Human MxA reduces the infection of DNA and RNA viruses, such as IBDV and ASFV		[68,69]
Viperin	RNA viruses	Targeting NS3 for proteasomal degradation		[48,49]
Tetherin		Inhibiting the replication of enveloped viruses		[50,51]
CH25H		Inhibiting the replication of coronaviruses		[52,53]
IFITM		Regulating the differentiation of CD4^+^ T cells and inhibiting viral infections		[54,55,56]
TRIM22/25		An E3 ubiquitin ligase		[57,58,59]
RLRs	RNA viruses and the IFN signaling pathways	Activating IFN transcription through the RIG-Ι/MDA-5 signaling pathway	Effectors of antiviral and positively regulated IFNs	[60,61,62,63,64]
OAS/RNase L		Degrading viral ssRNA		[47]
cGAS	DNA viruses and the IFN signaling pathways	Receptor for DNA viruses, induces type I IFNs and other cytokines		[44,45]
ALRs		Recognizing intracellular or nuclear pathogenic DNA or abnormal cellular DNA		[46]
PKR	IFN signaling pathways	Degrading IκBα	Positive regulators of IFNs	[73]
STAT1/2		Key adaptors in the JAK-STAT signaling pathway		[24]
IRF1, 3, 7 and 9		IFN regulatory factors		[70,71,72]
TRIM26		Facilitating the production of dsRNA-mediated type I IFNs		[74]
		Degrading IRF3 in the nucleus	Negative regulators of IFNs	[81]
USP18		Inhibiting the activation of the IFN signaling pathway by blocking the interaction between JAK and the IFN receptor		[77,78,79,80]
SOCS		Inhibiting JAK-STAT signaling by binding to phosphorylated tyrosine residues		[75,76]

## 3. Inflammatory Response Signaling Pathways Activated in Response to Viral Infections

During viral infections, PRRs recognize intracellular PAMPs and activate inflammatory response pathways, including the NF-κB and inflammasome signaling pathways.

### 3.1. NF-κB Signaling Pathways

Different PRRs mediate different immune response processes, but in the end, they all initiate the transcription and expression of immune-related genes by activating the NF-κB signaling pathway, of which NF-κB/Rel is the most important transcriptional regulator. Specifically, NF-κB/Rel regulates the transcription and expression of a variety of genes and is closely related to processes such as cell activation, cell proliferation, and immune and inflammatory responses. The NF-κB signal transduction pathways are divided into canonical and noncanonical pathways [82,83,84].

#### 3.1.1. Canonical Pathway

After recognition of the corresponding ligands by Toll-like receptors (TLRs), tumor necrosis factor (TNF) receptor (TNFR) and other receptors, the toll-interleukin-1 receptor (TIR) domain of the receptor is activated, which recruits myeloid differentiation factor 88 (MyD88) [85], TNF receptor-associated factor (TRAF) [86], phosphatidylinositide 3-kinase (PI3K) [87] or other mediators to the receptor and activates them. The signal transduction via MyD88 requires the interaction of TIR homodimers and IL-1 receptor-associated kinase (IRAK) [88], which phosphorylates MyD88 and recruits and activates TRAF6, followed by the activation of transforming growth factor β-activated kinase 1 (TAK1). The activated TAK1 leads to the eventual activation of the NF-κB signaling pathway and can activate activating protein (AP)-1, *c*-Jun N-terminal kinase (JNK), and p38. Upon activation, NF-κB and AP-1 are transported through nuclear pores to the nucleus [89], where they bind to promoters of specific genes to initiate their transcription and expression [90].

After binding to the corresponding PRR ligands, TNFR, T cell receptor (TCR), or B-cell receptor (BCR) recruits and activates the mediators to transfer signals downstream, activating transforming growth factor (TGF)-β, which activates TAK1. Activated TAK1 induces the formation of the IKK trimeric complex, which consists of catalytic (IKKα and IKKβ) and regulatory subunits (IKKγ, also known as NEMO) [91,92], and the IKK complex is activated and promotes the phosphorylation of IκBα, which is subsequently ubiquitinated and degraded in the presence of E3 ligase. Upon degradation, IκBα releases p50/RelA into the nucleus, which binds to the promoters of specific genes, initiating the transcription and expression of the genes encoding the proinflammatory cytokines TNF-α, IL-1α, IL-1β, IL-6, IL-10, cyclooxygenase-2 (COX-2), chemokine ligand (CCL)5, CCL19, CCL20, CC128, eotaxin, etc. [93] (Figure 3).

#### 3.1.2. Noncanonical Pathway

In contrast to the canonical pathway, the noncanonical pathway is dependent on IKKα but not NEMO [94]. The formation of p52/RelB heterodimers and dispensability of NEMO are hallmarks of the noncanonical pathway. When a cell receptor binds the corresponding ligand, it recruits and activates the adaptor NF-κB-inducing kinase (NIK), which activates IKKα. Subsequently, the activated IKKα binds p100 or p105, which is ubiquitinated and degraded by E3 ligase. p100 is degraded to generate p52, which can form a p52-RelB heterodimer or a p52-p52 homodimer. The p52/RelB heterodimer can enter the nucleus directly and bind to the promoter of a specific gene to initiate gene transcription and expression, while the p52/p52 homodimer forms a complex with B-cell lymphoma-3 (Bcl-3) and enters the nucleus to regulate the transcription and expression of a target gene [95]. The activation of noncanonical pathways contributes to the pathogenesis of various autoimmune and inflammatory diseases [96].

A central step in noncanonical NF-κB signaling is the stabilization of the NF-κB-inducing kinase (NIK) [97] and the downstream kinase IKKα, which induces phosphorylation-dependent processing of p100, an NF-κB precursor that also functions as a cytoplasmic inhibitor of NF-κB [98,99,100]. When RNA viruses (vesicular stomatitis virus and Sendai virus) infect the host cells, the NIK can negatively regulate type I IFN induction by viruses and TLR ligands via activation of noncanonical NF-κB signaling. Several studies have shown that negative regulation of type I IFN gene induction requires both of the noncanonical NF-κB members p52 and RelB. The canonical NF-κB member RelA potently synergizes with IRF3 in the induction of IFN-β in HEK293T cells. In contrast, the expression of p52 or RelB leads to the inhibition of the RelA-IRF3-stimulated IFN-β promoter activation [101].

At the same time, the noncanonical NF-κB members may compete with RelA for binding to the κB site on the IFN-β promoter. Altogether, virus-induced NIK upregulation and noncanonical NF-κB activation probably contribute to the suppression of type I IFN induction (Figure 3).

### 3.2. Inflammasome Signaling Pathways

The inflammasome is composed of receptors, ASC, and procaspase-1. The receptors are divided into nucleotide-binding and oligomerization domain (NOD)-like receptors (NLRs), ALRs, RLRs, and TLRs. ALRs usually recognize double-stranded viral DNA in the cytoplasm [46], RLRs recognize double-stranded viral RNA and 5′pppRNA in the cytoplasm [102,103], and TLR1, TLR2, and TLR4 localized on the cell surface detect viral glycoproteins and various bacterial PAMPs. TLR3, TLR7, and TLR9 localized on endosomes specifically detect viral or bacterial nucleic acids [104]. NLRP3 is triggered through the NF-κB signaling pathway [105,106], and ASC contains the CARD, which activates the inflammasome. Activation of the inflammasome induces the release of proinflammatory cytokines and pyroptosis.

The hallmark of the canonical inflammasome activation pathway is the cleavage and release of procaspase-1, proIL-1β, and proIL-18. Viral DNA in the cytoplasm is recognized by ALRs, and the cytoplasmic PAMPs produced by bacteria are recognized by NLR receptors [107]. Specifically, the activation of ALR or NLR receptors recruits the splice protein ASC to the downstream molecule procaspase-1, and the assembly of a multimeric protein complex that cleaves the CARD domain of procaspase-1 to produce caspase-1, which then recruits the downstream molecules proIL-1β and proIL-18 modulated by the NF-κB signaling pathway. The active subunits p20 and p10 of caspase-1 cleave proIL-1β and proIL-18 to generate the mature IL-1β and IL-18. In addition, caspase-1 can cleave the *N*-terminal structural domain of gasdermin D (GSDMD) [108,109,110], which is released and binds to the cellular membrane, where it forms pore channels, thus facilitating the secretion of mature IL-1β and IL-18 into the cell supernatants. In addition, extracellular water can enter the cell through GSDMD-generated pore channels, increasing the cell osmotic pressure, and cell contents can move out of these channels, causing pyroptosis [111,112] (Figure 1). A recent study revealed GSDMD *N*-terminal fragment displays a negative feedback effect to inhibit inflammasome-mediated activation of caspase-1/11 and relieve the excessive and uncontrolled inflammatory responses [113].

## 4. Crosstalk between IFN and Inflammatory Response Pathways in Response to Viral Infections

The IFN, NF-κB, and inflammasome signaling pathways play critical roles in antiviral immunity. When the inflammatory response pathway is activated, proinflammatory cytokines are released to cause pyroptosis in infected cells. At the same time, several important adaptors have been observed in different signaling pathways, and the complex interplay can play a positive or negative role in regulating the host’s antiviral immunity. On the one hand, viral infection causes an imbalance in host homeostasis, and IFN and inflammatory signaling pathways cooperate to eliminate viruses and maintain host homeostasis; on the other hand, there are antagonistic effects between IFN and the inflammatory response that inhibit the excessive immune response to facilitate host homeostasis.

### 4.1. Crosstalk between the NF-κB and IFN Signaling Pathways

It is generally accepted that crosstalk occurred among the NF-κB and IFN signaling pathways. The NF-κB/RelA protein binds to the enhancer of an IFN promoter and promotes the activation of the IFN pathway. The transcriptional process is triggered when TLR3 recognizes a ligand and thus recruits and activates TIR domain-containing adapter-inducing IFN-β (TRIF) [114], which activates TBK and protein kinase B (PKB, also known as Akt), ultimately inducing the activation of NF-κB/RelA. The activated NF-κB/RelA enters the nucleus through the nuclear membrane gap and binds to the enhancer of an IFN promoter, enhancing the transcription and expression of the *ifna* or *ifnb* gene. Classical swine fever virus (CSFV) infection triggers the RIG-I/MDA5-dependent signaling pathways, which in turn promote the entry of the transcription factors IRF3 and NF-κB/RelA into the nucleus to promote the secretion of IFN and inflammatory cytokines in porcine alveolar macrophages (PAMs) [115]. During porcine reproductive and respiratory syndrome virus (PRRSV) infection, the viral nonstructural protein NSP4 activates the NF-κB signaling pathway by reducing the phosphorylation-induced degradation of the IκBα factor and translocating RelA into the nucleus, thus downregulating the expression level of IFN-β in PAMs [116].

In addition, IRF2 enhances the activation of MyD88 and the TRIF-mediated NF-κB signaling pathway in carp. However, IRF2 is upregulated upon dsRNA virus infection, which negatively regulates the immune response triggered by the IFN signaling pathway and leads to an imbalanced immune response and facilitates the virus replication in carp [117].

As key enzymatic components of the NF-κB pathway, IKKα and IKKβ mediate crosstalk with components of other signaling pathways, including those in the p53, MAP kinase (MAPK), and IRF pathways, and directly regulate the transcription of ISGs in responses to viral infections [118]. IFN-mediated expression of PKR, an ISG protein, leads to a significant reduction in host cell translation and IκB degradation, which induces activation of the NF-κB signaling pathway (Figure 4). During the early stage of PRRSV infection, PKR overexpression activates NF-κB and IFN responses, which in turn enhances the expression of type I IFNs and proinflammatory cytokines, resulting in the reduction of PRRSV replication [119]. In addition, PKR may activate the NF-κB signaling pathway through interaction with the IKK complex in the Huh7 cells infected with yellow fever virus, thereby enhancing the antiviral effect of the pathway [120].

During viral infections, the IFN signaling pathway mainly regulates the NF-κB signaling pathway by mediating ISG expression, and the NF-κB signaling pathway significantly affects the activation of the IFN pathway through NF-κB/RelA. Therefore, the IFN and NF-κB signaling pathways can communicate with each other through some key adaptors, either synergistic or antagonistic, to play a positive or negative regulatory role in the antiviral responses.

### 4.2. Crosstalk between the Inflammasome and IFN Signaling Pathways

cGAS is a nuclear and cytosolic protein that responds to cytosolic dsDNA and catalyzes the formation of cyclic GMP-AMP (cGAMP), a second messenger that initiates the innate immune response via STING [121]. Caspase-1 has been shown to play a central role in the relationship between type I IFNs and inflammasome activation [122]. Caspase-1 is activated upon inflammasome activation to cleave cGAS [123], thus inhibiting the IFN pathway upon DNA viral infection (Figure 5). For example, herpes simplex virus 1 (HSV-1) infection inhibits IFN-β release by enhancing caspase-1 activation in stratified squamous epithelial cells [124]. Thus, caspase-1 exerts a negative effect on the activation of IFN during DNA virus infection, and caspase-1 may suppress autoimmunity induced by the IFN pathway.

The ZIKV NS1 protein induces activation of the NLRP3 inflammasome to cleave cGAS and further evade antiviral response [126]. Influenza A viruses (IAVs) have evolved various strategies to counteract innate immune responses through viral proteins. In mammalian cells, the phox and bem 1 (PB1) protein of the avian influenza A (H7N9) virus promotes ring finger protein 5 (RNF5) to catalyze the polyubiquitination of lysine (K)27-linked MAVS at K362 and K461 to inhibit host type I IFN production [127]. Furthermore, the PB1-F2 protein selectively antagonizes RNA-induced NLRP3 inflammasome activation by inhibiting MAVS-NLRP3 crosstalk [128]. Therefore, due to the interactive characteristics of innate immune signaling pathways, viral proteins can affect not only innate immune adaptors but also the interacting partners (Figure 6).

MAVS can promote NLRP3 inflammasome signaling to enhance the activation of caspase-1 and the production of downstream molecules of the NLRP3 inflammasome (Figure 6). In addition, MAVS can promote the recruitment of NLRP3 to mitochondria. Mitochondrial reactive oxygen species (mtROS) and mitochondrial DNA (mtDNA) act as primary damage-associated molecular patterns (DAMPs) to promote NLRP3 oligomerization and activation in human cells [129]. Therefore, MAVS promotes the recruitment of NLRP3 to mitochondria. Furthermore, the oligomerization and activation of NLRP3 can be enhanced by mitochondrial reactive oxygen species (ROS). SeV infection induces the activation of MAVS signaling by promoting NLRP3-dependent caspase-1 activation, whereas knockdown of MAVS expression significantly attenuated the NLRP3 inflammasome in human acute monocytic leukemia cells (THP-1) and mouse macrophages [130].

Notably, mtDNA can also act as a potent cGAS agonist, promoting activation of the cGAS-STING signaling pathway. Briefly, mitochondrial outer membrane permeabilization (MOMP) or other forms of mitochondrial dysfunction result in mtDNA release to the cytosol and induce the activation of cGAS signaling in mice [131,132,133]. There is a multi-layer crosstalk between the inflammatory and IFN pathways, which are involved in adaptors in innate signaling pathways or various organelles. Therefore, multiple innate immune signaling pathways orchestrate each other to maintain the homeostasis of host cells.

### 4.3. Common Adaptors in the Regulation of IFN, NF-κB, and Inflammasome Signaling Pathways

NF-κB activation and type I IFN signaling must be tightly regulated to maintain host homeostasis, and excessive or uncontrolled immune responses cause extensive tissue injury [134].

NEMO-like kinase (NLK), a host enzyme, modulates the balance of the IFN and NF-κB signaling pathways to maintain host homeostasis during viral infections. It has been shown that NLK inhibits the virus-induced activation of the type I IFN signaling in a mouse model [135]. Meanwhile, NLK regulates NF-κB signaling by disrupting the interaction between TAK1 and IKK in HEK293 cells [136] (Figure 4).

IRF3 and IRF9 are closely associated with *ifn* gene expression during viral infection. When a virus infects cells, the adaptors MAVS, STING, and TRIF activate the downstream protein kinase TBK1, leading to the phosphorylation of the transcription factor IRF3, thereby promoting the production of type I IFNs [137]. During the process of TBK1-mediated activation of IRF3, MAVS, and STING, which contain two conserved serine and threonine clusters, are phosphorylated by IKK and TBK1 upon stimulation. Phosphorylated MAVS and STING then bind to the positively charged surface of IRF3, thereby recruiting IRF3, which is phosphorylated and activated by TBK1 [138]. The IRF3 of *Scleractinian* species can activate the NF-κB signaling pathway by enhancing the degradation of IκBα, thereby facilitating the antiviral response of the host [139]. In addition, IRF9 overexpressed in *Scleractinian* can exert a negative feedback or inhibitory effect on the TRIF-mediated NF-κB signaling pathway through the IRF9 DNA-binding domains (DBD) [140].

The host antiviral response is a sophisticated and interactive signaling regulatory network. Several cellular proteins participated in the crosstalk between the IFN and NF-κB pathways, suggesting that the IFN and NF-κB pathways coordinate with each other to exert antiviral activities and maintain homeostasis during antiviral infection. The ovarian tumor family deubiquitinase 4 (OTUD4) targets MAVS to remove K48-linked polyubiquitin chains upon RNA virus infections, thereby maintaining MAVS stability and facilitating the signal transduction in a mouse model [141] (Figure 6). In addition, OTUD4 can also target and deubiquitinate MyD88, an important coupling molecule in the NF-κB signaling pathway, thereby inhibiting the activation of the NF-κB signaling pathway [142]. Similarly, OTUD4 plays a similar role in the recruitment of MyD88 to TRAF6, which inhibits the activation of the NF-κB signaling pathway by inhibiting the K63-linked ubiquitination of TRAF6 to relieve the inflammatory responses [143] (Figure 4).

## 5. Concluding Remarks and Prospects

More and more information on the regulation of innate immunity upon viral infections has been deciphered. Importantly, interplays and crosstalk in innate immune signaling pathways are ubiquitous and constitute sophisticated regulatory networks. However, research into crosstalk in innate immune signaling pathways is still in its infancy. It is generally accepted that an excessive inflammatory response induced by viral infection can influence other innate immune responses and further disturb host homeostasis. SARS-CoV2 infection impairs the imbalance of innate immunity, which results in the “cytokine storm” and causes severe illness [144,145]. ZIKV infection triggers the activation of the inflammasomes and further inhibits IFN production, thus attenuating the antiviral effects induced by IFN. Therefore, it is important to identify the regulators responsible for maintaining the balance between IFNs and inflammatory activation. Inflammasome activation triggers caspase-1-mediated cleavage of cGAS to inhibit the type I IFN response during DNA virus infection, and some RNA viruses, such as DENV, can also activate the cGAS/STING pathway, but the mechanism of caspase-1-mediated cleavage of cGAS and other adaptors to inhibit the type I IFN response during RNA virus infection remains unclear. Furthermore, the crosstalk between innate immunity and other signaling pathways, including cell stress and autophagy, remains largely unknown. In addition, the process of signal transduction is accompanied by the transformation of matter and energy, and it is likely that metabolism participates in the regulation of the crosstalk of innate immune signaling pathways. Several important issues described above need to be further investigated. Increasing knowledge of crosstalk will contribute to a comprehensive understanding of the complex interplay between viruses and hosts and provide molecular targets for the development of vaccines or antivirals in the future.

## Figures and Tables

**Figure 1 viruses-14-02798-f001:**
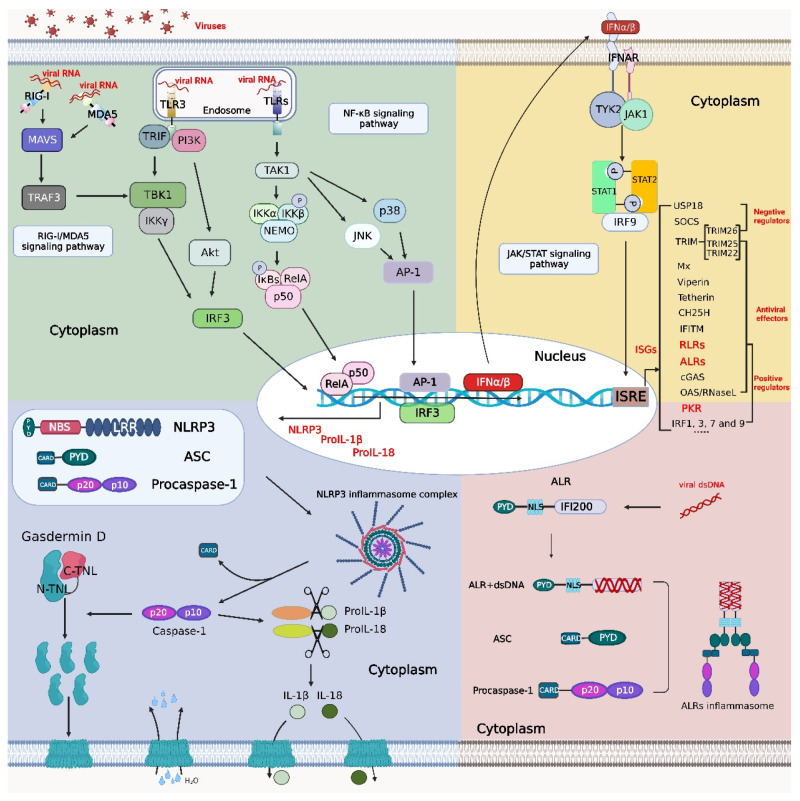
Crosstalk among the NF-κB, inflammasome, and IFN signaling pathways. Viral RNA binds to retinoic acid-inducible gene-I (RIG-I)-like receptors (RLRs) and Toll-like receptors (TLRs) to activate the production of type I interferon (IFN), nucleotide-binding and oligomerization domain (NOD)-like receptor protein 3 (NLRP3), pro-interleukin (IL)-1β and pro-IL-18 through the RIG-I/melanoma differentiation-associated gene 5 (MDA5) signaling pathway and nuclear factor kappa B (NF-κB) signaling pathway, respectively. After recognition of type I IFNs by the receptor interferon-α/β receptor (IFNAR), a signal is transmitted downstream to initiate the Janus kinase (JAK)/signal transducer and activator of transcription (STAT) signaling pathway. STAT1, STAT2, and interferon regulatory factor (IRF) 9 form the interferon-stimulated gene factor (ISGF) 3 complex, which enters the nucleus to initiate the transcription of IFN-stimulated genes (ISGs) such as protein kinase R (PKR), IRFs, RLRs, absent in melanoma 2 (AIM2)-like receptors (ALRs), etc. ALRs can recognize double-stranded DNA (dsDNA) to initiate inflammasome signaling pathways, and PKR displays a degradation effect on inhibitor of NF-κB (IκB) (a key regulator in the NF-κB signaling pathway). Activation of the NF-κB signaling pathway promotes the synthesis of NLRP3, pro-IL-1β, and pro-IL-18 and facilitates the assembly of the NLRP3 inflammasome complex. The figure was created using the online software “Biorender” (https://app.biorender.com/ The updated figure was created on 30 November 2022).

**Figure 2 viruses-14-02798-f002:**
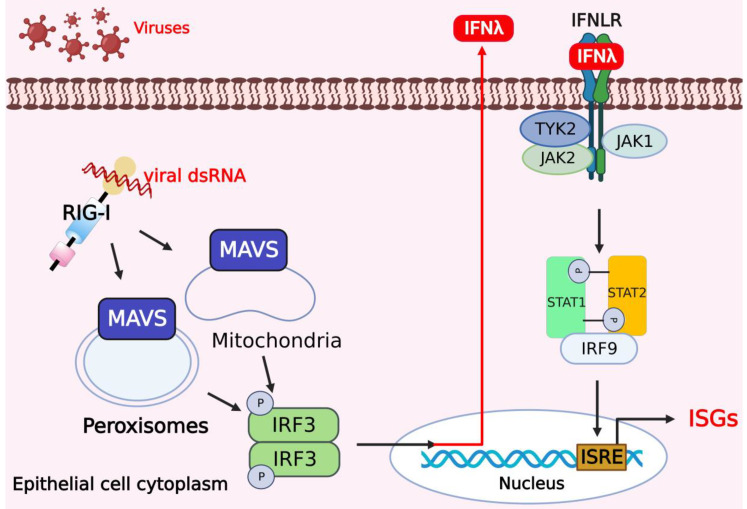
Activation of type III IFNs and downstream signaling pathways in response to viral infections. In human cells, especially in epithelial cells, mitochondrial antiviral signaling protein (MAVS) on peroxisomes and mitochondria promotes the expression of IFN-λ to activate the JAK/STAT pathway regulated by JAK2 and induces selective expression of IFN-λ. The figure was created using the online software “Biorender” (https://app.biorender.com/ The updated figure was created on 7 December 2022).

**Figure 3 viruses-14-02798-f003:**
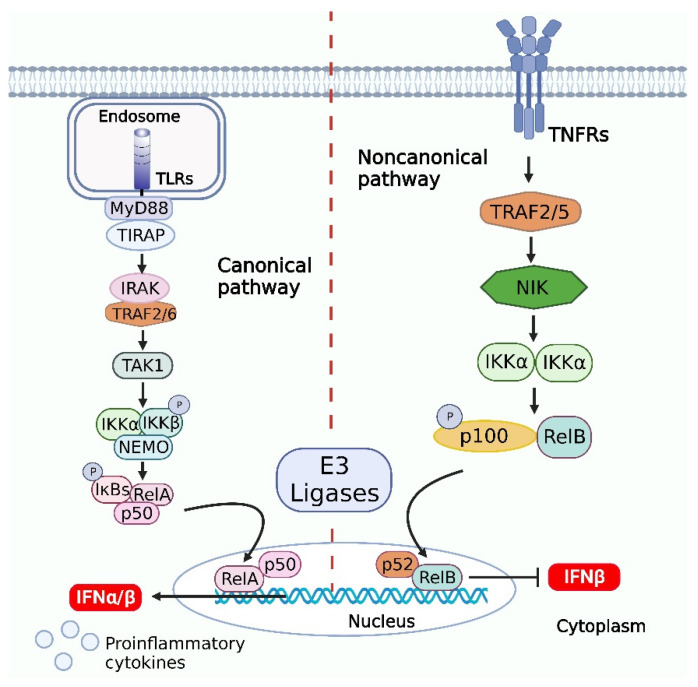
Activation of NF-κB signaling pathways and downstream signaling pathways in response to viral infections. Canonical pathway, the signal transduction via myeloid differentiation factor 88 (MyD88) requires the interaction of toll-interleukin-1 receptor (TIR) homodimers and IL-1 receptor-associated kinase (IRAK), which phosphorylates MyD88 and recruits and activates tumor necrosis factor (TNF) receptor-associated factor (TRAF)6, followed by the activation of transforming growth factor β-activated kinase (TAK)1. The activated TAK1 induces the formation of the inhibitor of NF-κB kinase trimer (IKK) trimeric complex, and the IKK complex is activated and promotes the phosphorylation of IκBα, which is subsequently ubiquitinated and degraded in the presence of E3 ligase. Upon degradation, IκBα releases p50/RelA into the nucleus, which binds to the promoters of specific genes, initiating the transcription and expression of the genes. The noncanonical pathway is dependent on IKKα but not NF-κB essential modulator (NEMO), and when a cell receptor binds to the corresponding ligand, it recruits and activates the adaptor NF-κB-inducing kinase (NIK), which activates IKKα. Subsequently, the activated IKKα binds to p100 or p105, which is ubiquitinated and degraded by E3 ligase. p100 is degraded to generate p52, which can form a p52-RelB heterodimer or a p52-p52 homodimer. The p52/RelB heterodimer can enter the nucleus directly and bind to the promoter of a specific gene to initiate gene transcription. In addition, it is likely that the virus-induced NIK upregulation and noncanonical NF-κB activation inhibit the type I IFN induction. The figure was created using the online software “Biorender” (https://app.biorender.com/ The updated figure was created on 7 December 2022).

**Figure 4 viruses-14-02798-f004:**
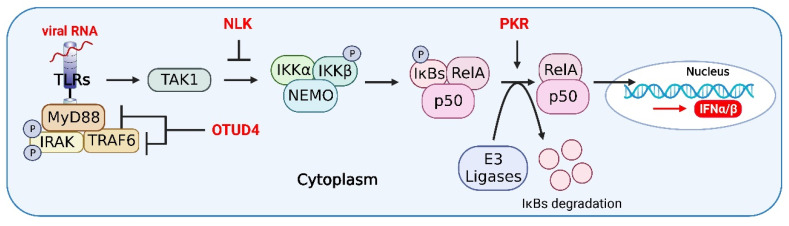
Host proteins in the crosstalk of IFN, NF-κB, and inflammasome signaling pathways. OTUD4 antagonizes the recruitment of MyD88 to TRAF6, which inhibits the activation of the NF-κB signaling pathway by inhibiting the K63-linked ubiquitination of TRAF6, thereby relieving inflammatory responses in host cells. NEMO-like kinase (NLK) regulates NF-κB signaling by disrupting the interaction between TAK1 and IKK. PKR leads to a significant reduction in host cell translation and IκB degradation, which induces activation of the NF-κB signaling pathway. The figure was created using the online software “Biorender” (https://app.biorender.com/ The updated figure was created on 30 November 2022).

**Figure 5 viruses-14-02798-f005:**
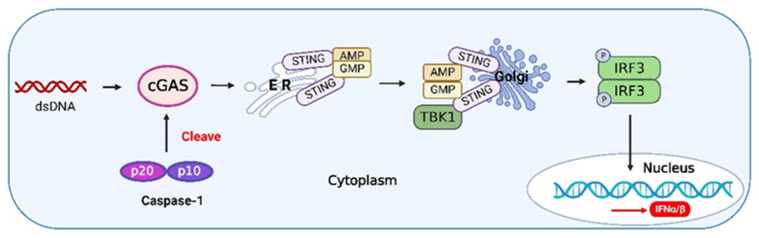
The crosstalk between caspase-1 and type I IFNs. Upon binding to DNA, cyclic GMP-AMP synthase (cGAS) undergoes a conformational change to generate an active state and produces the second messenger cyclic GMP-AMP (cGAMP) from adenosine triphosphate (ATP) and guanosine triphosphate (GTP), which is subsequently detected by the cyclic-dinucleotide sensor (CDNS), stimulator of interferon genes (STING), a transmembrane protein located in the endoplasmic reticulum (ER). Binding of cGAMP induces the activation of STING, and the activated STING is translocated to the Golgi and activates tumor necrosis factor (TNF) receptor-associated factor (TRAF) family member-associated NF-κB activator (TANK)-binding kinase 1 (TBK1). TBK1 phosphorylates IRF3 and induces its dimerization. The dimerized IRF3 enters the nucleus and triggers the production of type I IFNs [125]. Activated caspase-1 during inflammasome activation can cleave cGAS, inhibiting the IFN pathway induced by DNA viruses. The figure was created using the online software “Biorender” (https://app.biorender.com/ The updated figure was created on 30 November 2022).

**Figure 6 viruses-14-02798-f006:**
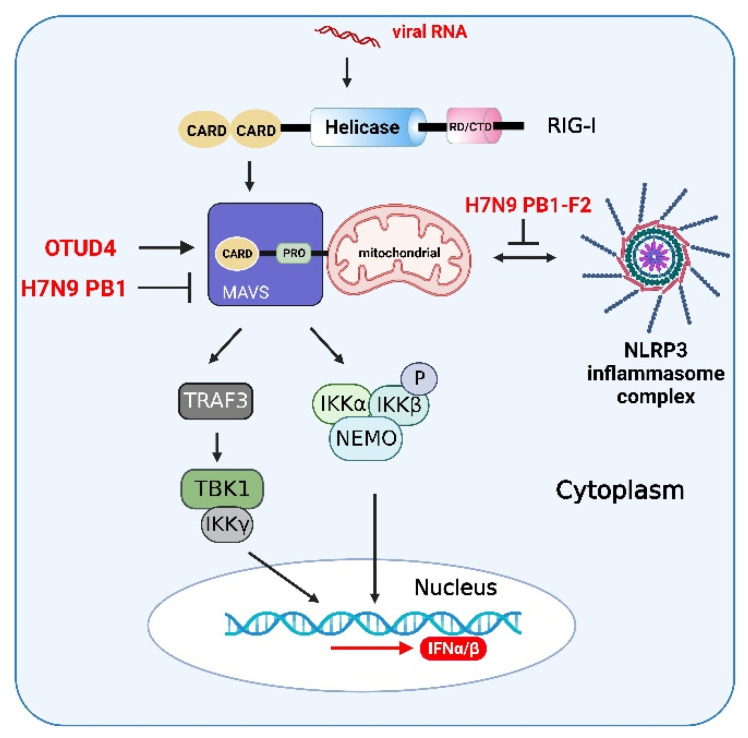
Viral proteins and host proteins in the crosstalk of IFN, NF-κB, and inflammasome signaling pathways. After activation of RIG-I-mediated signaling pathway induced by RNA virus infections, ovarian tumor family deubiquitinase 4 (OTUD4) targets MAVS to remove the Lys (K)48-linked polyubiquitin chains, thereby maintaining MAVS stability and promoting innate antiviral signaling. The PB1 protein of the H7N9 virus promotes RNF5 to catalyze the K27-linked polyubiquitination of MAVS at K362 and K461 to inhibit host type I interferon production. The crosstalk between MAVS and NLRP3 and the PB1-F2 protein selectively antagonize RNA-induced NLRP3 inflammasome activation by inhibiting MAVS-NLRP3 crosstalk. The figure was created using the online software “Biorender” (https://app.biorender.com/ The updated figure was created on 30 November 2022).

## Data Availability

All data generated or analyzed during this study are included in this published article.

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
