# Peer review of "New Insights into the Crosstalk among the Interferon and Inflammatory Signaling Pathways in Response to Viral Infections: Defense or Homeostasis"

_viruses, 2022, doi:10.3390/v14122798_

Round 1

Reviewer 1 Report

Dai et al., have submitted a comprehensive review entitled "New insights into the crosstalk among interferon and inflammatory signaling pathways in response to viral infections: Defence or homeostasis".  

The authors well explained the innate sensors on antiviral response, particularly interferon and inflammatory signaling pathways. 

Over the review gets good credit. However, minor changes are necessary to improve the quality of the manuscript. 

Introduction: Second paragraph, "In addition.....signaling." should be rewritten for clarity.

Page 2: Second line, punctuation error. 

Expand and define some terminology at first instance., e.g., PKT, ALRs, Mx, etc. 

Figure 1. The arrow from Caspase 1 should show directly on the pro-IL-1b and pro-IL-18.

The authors have discussed the different forms of IFNs; however, the figure represents only type I; why so?

Like positive regulators, the negative regulators should be discussed in detail. 

Reviewer 2 Report

Dear authors,

Your manuscript “New insights into the crosstalk among interferon and inflammatory signaling pathways in response to viral infections: Defense or homeostasis” is very interesting to read and gives a good review on anti-virus response mechanisms.

But some issues should be resolved:

First, some corrections in English are required.

Second, I have the following remarks on the manuscript:

Page 2 – “PKR” – what is it? Has to be specified.

Page 3 – “PAMPs” – what is it? Has to be specified.

To clarify – were manuscript figures drawn using some kind of specialized software/service? If so, please indicate it.

Reviewer 3 Report

This review provides a broad summary of the inflammatory signaling pathways activated by viral infections. Comments on how to improve the manuscript are provided below:

Section 2.1, first line: IFNs are secreted into cell supernatant in vitro, but in vivo this would be secreted into the extracellular matrix. This sections needs to clarify whether these studies have been conducted in vitro and/or in vivo and ensure the correct terminology is used.

In section 2.1, more detail should be provided on type III IFN signaling, rather than just saying that type III IFNs activate the JAK/STAT pathway. The previous section describes two different JAK/STAT pathways (for type I and II IFNs), so the signaling that occurs for IFN lambda should be described.

It would be good to also mention that IFNs signal through non-canonical signaling pathways, especially because this is done for NF-kB.

Section 2.2.1 has a lot of good information. A table at the end of this section summarizing the ISGs, their effects, and the viruses they target would be useful in summarizing the information for the reader. This could also be done for the information in sections 2.2.2 and 2.2.3.

Section 3 on NF-kB signaling pathways. A diagram showing these pathways would be a useful accompaniment to the text.

Several examples are given throughout the manuscript of viruses inducing inflammatory effects or activating different signaling pathways. But details are lacking on how these studies were conducted eg. the cell type, whether it occurs in vitro or in vivo, mouse or human etc. This is very important information to include because the immune response to viruses can greatly differ between cell types in the body, and between different species.

Make sure all abbreviations are defined the first time they are mentioned. eg. CARD is not defined until the 3rd time it is mentioned

Make sure all statements are referenced. For eg. in section 2.1, the sentence "the receptor of type III IFNs is a heterodimer consisting of IL-10R2 and IL-28Ra, and it is expressed only on epithelial cells" is an unreferenced statement. There are several examples of this throughout the manuscript that should be corrected. It's also worth noting that IFNLRs are mostly expressed on epithelial cells, but there is evidence in the literature that this receptor is expressed on other cell types as well (eg. immune cells)

Round 2

Reviewer 3 Report

The authors have addressed all the concerns raised during the first review.